Reverse transcription recombinase polymerase amplification-lateral flow assay for detection of pathogenic orthoflaviviruses in mosquito vectors

Thayanukul Parinda Parinda.tha@mahidol.ac.th 1 2
Morales Vargas Ronald Enrique 2 3
Sujijun Konkamon 1 2
Khumpeera Pimchanok 1
Suksawat Kittiya 1
Wijegunawardana Nahallage Dona Asha Dilrukshi 4
Rijiravanich Patsamon 5 6
Surareungchai Werasak 7
Kittayapong Pattamaporn 2
1 Department of Biology, Faculty of Science, Mahidol University , Ratchathewi , Bangkok , Thailand
2 Center of Excellence for Vectors and Vector-Borne Diseases, Faculty of Science, Mahidol University , Salaya , Nakhon Pathom , Thailand
3 Department of Pharmacology, Faculty of Science, Mahidol University , Ratchathewi , Bangkok , Thailand
4 Department of Bioprocess Technology, Faculty of Technology, Rajarata University of Sri Lanka , Mihintale , Sri Lanka
5 Industrial Sensor Technology Research Team, National Center for Genetic Engineering and Biotechnology, National Sciences and Technology Development Agency at King Mongkut’s University of Technology Thonburi , Bang Khun Thian , Bangkok , Thailand
6 Sensor Technology Laboratory, Pilot Plant Development and Training Institute, King Mongkut’s University of Technology Thonburi , Bang Khun Thian , Bangkok , Thailand
7 Nanoscience & Nanotechnology Graduate Programme, Faculty of Science, King Mongkut’s University of Technology Thonburi , Bang Mot , Bangkok , Thailand
Brygadyrenko Viktor
Electronic publication date: 2025 Aug 26
Publication date: 2025
Volume: 13
Electronic Location ID: e19820
Received 2024 Sep 23; Accepted 2025 Jul 10
Copyright: ©2025 Thayanukul et al.
Copyright year: 2025
Copyright holder: Thayanukul et al.
License: This is an open access article distributed under the terms of the Creative Commons Attribution License, which permits unrestricted use, distribution, reproduction and adaptation in any medium and for any purpose provided that it is properly attributed. For attribution, the original author(s), title, publication source (PeerJ) and either DOI or URL of the article must be cited.
License URL: https://creativecommons.org/licenses/by/4.0/

Keywords: Orthoflavivirus, Reverse Transcription Recombinase Polymerase Amplification (RT-RPA), Lateral Flow Detection (LFD), Mosquito, Arbovirus, Recombinase Polymerase Amplification (RPA), Dip strip, Biosensor, Mosquito-borne diseases, Vector

Funding: Mahidol University National Science and Innovation Fund Center for Scientific Instrumentation and Platform Services, CIF and CNI Grant, Faculty of Science, Mahidol University This research project is supported by Mahidol University (Fundamental Fund: fiscal year 2023 by National Science and Innovation Fund (NSRF)) and the Center for Scientific Instrumentation and Platform Services, CIF and CNI Grant, Faculty of Science, Mahidol University. The funders had no role in study design, data collection and analysis, decision to publish, or preparation of the manuscript.

==============================
Background

The genus Orthoflavivirus primarily consists of arthropod-borne viruses capable of infecting vertebrate hosts and causing serious human diseases such as dengue fever, Zika fever, Japanese encephalitis, West Nile fever, and yellow fever. This study describes the development of a simple and field-deployable detection system for multiple pathogenic orthoflavivirus species using the recombinase polymerase amplification (RPA) technique.

Methods

Several previously published broad-specific primers targeting the genus Orthoflavivirus were evaluated. A new primer pair, FlaviPath-F and FlaviPath-R, was designed and tested for its applicability in an RPA assay. The RPA protocol was experimentally optimized, with a focus on determining the assay’s sensitivity and assessing the primers’ specificity against pathogenic orthoflaviviruses.

Results

The primer FlaviPath-F and FlaviPath-R targeted 36% of the selected pathogenic orthoflavivirus species without cross-reacting with non-pathogenic strains based on in silico analysis. The RPA assay successfully amplified DNA oligonucleotides from dengue virus, Japanese encephalitis virus, Zika virus, and West Nile virus. Furthermore, positive amplification was observed in RNA samples extracted from mosquitoes infected with dengue and Zika viruses. The RPA assay demonstrated high sensitivity, with the potential to detect as few as a single viral RNA copy, although confirmation is needed for concentrations below the detection limit of 104 RNA copies.

Discussion

This is the first study to develop an RPA-based method for the detection of multiple orthoflavivirus pathogens in mosquito vectors. The reverse transcription recombinase polymerase amplification assays with lateral flow dipsticks (RT-RPA-LFD) platform offers a rapid, cost-effective tool for identifying regions at risk of arboviral transmission, supporting the targeting of individual viral diseases. This technique holds promise as an early warning system for emerging arboviral threats in public health.

Introduction

Orthoflavivirus (formerly Flavivirus) is a genus within the Flaviviridae family, characterized by a positive-sense single-stranded RNA genome with a type I cap structure (m7GpppAmp) at the 5′-end (International Committee on Taxonomy of Viruses, 2024). The virion comprises three structural proteins—capsid, membrane, and envelope—and seven nonstructural proteins (NS1, NS2A, NS2B, NS3, NS4A, NS4B, and NS5). Most orthoflaviviruses are arthropod-borne, capable of infecting vertebrate hosts and causing severe human diseases. Some target the central nervous system—such as Zika virus (O. zikaense, ZIKV), Japanese encephalitis virus (O. japonicum, JEV), and West Nile virus (Orthoflavivirus nilense, WNV)—while others like dengue virus (O. dengue, DENV) and yellow fever virus (O. flavi, YFV) can lead to hemorrhagic fever and organ failure, respectively. In contrast, certain orthoflaviviruses, including Aedes flavivirus (AEFV), Nakiwogo virus (NAKV), and Lammi virus (LAMV), are known to infect only insects and have not been linked to human disease (Cook et al., 2009; Guzman et al., 2018; Kholodilov et al., 2024). Among common orthoflavivirus infections, DENV is recognized as the fastest-spreading mosquito-borne virus (Liang & Dai, 2024). In 2021, the estimated global cases of major pathogenic orthoflaviviruses included 58,964,185 cases of dengue, 169,734 cases of Zika virus, and 86,509 cases of yellow fever.

Studying viruses within their hosts is critical for identifying novel transmission pathways and establishing early-warning systems to mitigate potential outbreaks. Numerous detection methods targeting pathogenic orthoflaviviruses have been developed, with polymerase chain reaction (PCR)-based techniques being the most common. Several primer sets have been designed to target conserved regions within the orthoflavivirus genome (Bronzoni et al., 2005; Fulop et al., 1993; Tanaka, 1993), often incorporating degenerate bases to broaden viral detection coverage (Daidoji et al., 2021; Johnson et al., 2010; Kuno, 1998; Maher-Sturgess et al., 2008; Pierre, Drouet & Deubel, 1994; Scaramozzino et al., 2001a; Scaramozzino et al., 2001b; Vina-Rodriguez et al., 2017; Xue et al., 2021). However, PCR-based assays require rather expensive thermal cycler and may take hours to produce results (Kuno, 1998; Xue et al., 2021). In contrast, isothermal amplification combined with lateral flow detection (LFD), requiring only a heat block, offers a faster, simpler alternative suitable for use in resource-limited field settings.

Several isothermal amplification methods have emerged, including loop-mediated isothermal amplification (LAMP), recombinase polymerase amplification (RPA), nucleic acid sequence-based amplification (NASBA), helicase-dependent amplification (HDA), and strand displacement amplification (SDA). Among these, RPA is particularly advantageous due to its low operating temperatures (37–42 °C), with some applications feasible even at body temperature (axilla) (Crannell, Rohrman & Richards-Kortum, 2014). RPA is highly sensitive and specific, capable of detecting as few as 1–10 target DNA copies within 20 min (Lobato & O’Sullivan, 2018).

The RPA process begins with the formation of a complex between recombinase protein uvsX (derived from T4-like bacteriophages) and primers in the presence of ATP and a high molecular polyethylene glycol (crowding agent). This complex searched for a homologous sequence in the target DNA. Single-stranded binding proteins stabilize the unwound DNA, allowing strand-displacing DNA polymerase to initiate elongation in the presence of dNTPs. The cyclic nature of this mechanism lead to exponential amplification (Lobato & O’Sullivan, 2018; Stringer et al., 2018).

Most RPA-based studies for orthoflavivirus detection have focused on individual pathogens in human samples, such as DENV, ZIKV, YFV, and Tembusu virus (Abd El Wahed et al., 2015; Ahmed et al., 2022; Chan et al., 2016; Escadafal et al., 2014; Leon et al., 2022; Myhrvold et al., 2018; Teoh et al., 2013; Xi et al., 2019; Yin et al., 2022), often using LFD dipstick for result visualization. Some protocols have even bypassed RNA extraction, applying RPA directly to mosquito lysates for dengue virus detection (Ahmed et al., 2022; Bonnet, van Jaarsveldt & Burt, 2022). However, few studies have explored multiple detection of orthoflaviviruses. One approach used dual probe sets for reverse transcription recombinase polymerase amplification (RT-RPA)-based detection (reverse transcriptase-RPA) of WNV and Wesselsbron virus (WSLV) via LFD (Bonnet, van Jaarsveldt & Burt, 2022). Another study integrated the clustered regularly interspaced short palindromic repeats (CRISPR-Cas13) system with RT-RPA to improve detection specificity and sensitivity for DENV, ZIKV, WNV, and YFV. This method used a group-specific primer set paired with species-specific CRISPR RNAs, enabling simultaneous detection of one or two viruses using a single fluorescent wavelength in a multi-well plate format (Myhrvold et al., 2018). More recently, flap endonuclease 1 (FEN1)-aided RPA (FARPA) was employed to simultaneously detect ZIKV, tick-borne encephalitis virus, and YFV through reactions with various fluorescent probes, with visualization via a real-time PCR system or a handheld FARPA analyzer (Ma et al., 2023).

In this study, we present the development of a simple, largely instrument-free RPA-based platform for on-site detection of multiple pathogenic orthoflaviviruses. To broaden detection coverage, primers were designed with degenerative bases to target conserved genomic regions. The assay is intended for use on mosquito samples and selectively detects pathogenic orthoflaviviruses while excluding non-pathogenic viruses. This method has the potential to serve as both a field-deployable early warning system for public health surveillance and a valuable tool for exploring novel or understudied host-vector transmission dynamics of emerging orthoflaviviruses.

Materials & Methods

Ethics statement

The use of mosquito specimens in this study was approved by the Faculty of Science, Mahidol University Animal Care and Use Committee (SCMU-ACUC) under protocol number MUSC66-020-650. All experiments were conducted at the Center of Excellence for Vectors and Vector-Borne Diseases (CVVD), Faculty of Science, Mahidol University.

Recombinase polymerase amplification (RPA) primer design

Viral genome sequences of the genus Orthoflavivirus were retrieved from the International Committee on Taxonomy of Viruses Database (ICTV, https://ictv.global) and the National Center for Biotechnology Information (NCBI, https://www.ncbi.nlm.nih.gov). DNA sequences were obtained between June 2023 and June 2024. Detailed information of each sequence, including sample collection dates, is provided in Table S4. Sequence alignments were performed using Geneious Prime software (GraphPad Software LLC, Auckland, New Zealand) with the Clustal Omega algorithm under the “Group sequences by similarity” mode. Assignment of human pathogenic and non-pathogenic sequences was manually performed by reviewing relevant literature. Primers broadly targeting Orthoflavivirus were obtained from previous PCR studies (Table S3; (Bonnet, van Jaarsveldt & Burt, 2022; Bronzoni et al., 2005; Daidoji et al., 2021; Fulop et al., 1993; Grubaugh et al., 2013; Johnson et al., 2010; Kuno, 1998; Maher-Sturgess et al., 2008; Myhrvold et al., 2018; Patel et al., 2013; Pierre, Drouet & Deubel, 1994; Rice et al., 1985; Scaramozzino et al., 2001b; Tanaka, 1993; Vina-Rodriguez et al., 2017; Xue et al., 2021)) and assessed for binding ability to pathogenic Orthoflavivirus species, but not to non-pathogenic ones. Primers demonstrating the potential to bind across multiple pathogenic Orthoflavivirus species were shortlisted and served as the basis for designing new primers. These primers were further refined following the RPA design criteria outlined in the TwistAmp® DNA Amplification Kits Assay Design Manual (part number: INASDM Revision 1, https://www.twistdx.co.uk). The final primer sequences were: Flavipath-F (5′-AAR GGH AGY MGN GCH ATH TGG TWY ATG TGG-3′) and Flavipath-R (5′- CCT TCH ACW CCD CYB HVD GAR TTY TYH CKV-3′). Primers were synthesized by Macrogen (Seoul, Republic of Korea) using phosphoramidite chemistry at a 50 nmole scale and purified by MOPC (cartridge purification). Probes were synthesized by Synbio Technologies (USA) at 100 nmole and 2 OD yield, with HPLC purification. For the dipstick test, primers were labeled with Digoxigenin (DIG) and fluorescein isothiocyanate (FITC) corresponding to the antibody binding sites of the Amplicon Detection 2T1C Dipstick (K-BIOSCIENCES, Pathumthani, Thailand). Detailed primer and probe sequences are provided in Table S1.

DNA and RNA sample preparation

Synthetic DNA targets were synthesized using phosphoramidite chemistry and ordered from Macrogen (Seoul, Republic of Korea) either as single-stranded oligonucleotides or as double-stranded oligo duplexes. All were prepared at a 50 nmole scale and purified by MOPC (Table S2). RNA targets were synthesized using the MAXIscript™ T7 Transcription kit (Invitrogen, Waltham, MA, USA), treated with TURBO DNase1 and 0.5 M EDTA, and precipitated with ammonium acetate/ethanol according to the manufacturer’s protocol. cDNA templates were synthesized using the SuperScript™ III One-Step RT-PCR System with Platinum™ Taq DNA Polymerase (Invitrogen), utilizing modified primers T7-DENV2 NS5-F and DENV-2 NS5-R (Patramool et al., 2013; Richardson et al., 2006) under the following thermal conditions: 55 °C for 30 min, 94 °C for 2 min, followed by 40 cycles of 94 °C for 15 s, 60 °C for 15 s, and 72 °C for 1 min.

RNA samples were extracted from the whole bodies of Aedes aegypti mosquitoes infected with dengue virus serotype 2 (Thai strain ThNR02/772) or Zika virus (Brazilian strain BE H 815744) without dissection. Viruses were propagated and mosquito infections performed according to Morales-Vargas et al. (2020) and Pompon et al. (2017). Mosquitoes were stored for approximately three years at −80 °C freezer before processing (n = 3 for each virus). Individual mosquito was homogenized in 1xPBS for 100 µL using a TissueLyser LT (Qiagen, Hilden, Germany) or pellet pestles, followed by centrifugation at 12,000 rpm for 5 min at 4 °C. The supernatant was mixed with one mL of Trizol LS Reagent (Invitrogen, USA), and RNA was extracted according to the manufacturer’s instructions, including the optional centrifugation step for high-fat samples. RNA pellets were resuspended in 25 µL DEPC-treated water (Invitrogen) and incubated at 55 °C for 10 min. Samples were stored at −80 °C until use. All molecular works was performed on clean benches decontaminated with 70% EtOH and RNase AWAY (Thermo Fisher Scientific, Waltham, MA, USA).

Recombinase Polymerase Amplification (RPA) reaction

The RPA reaction was performed using the TwistAmp basic kit (TwistDx™ Limited, UK) following the manufacturer’s instructions with modifications. Modifications included splitting the master mix for multiple reactions and adding a termination step at 65 °C for 10 min (Kumar et al., 2022; Patchsung et al., 2020). The master mix consisted of: 29.5 µL rehydration buffer, 4.0 µL Nuclease-Free water (Invitrogen), 2.5 µL Flavipath-F probe (10 µM), and 2.5 µL Flavipath-R probe (10 µM). Four 9.6 µL aliquots were prepared: each mixed with 1 µL of DNA template or nuclease-free water (for the non-template control, NTC). MgAcO solution (0.29 µL) was added immediately before incubation. Reactions were incubated at 37 °C for 30 min or different temperatures as specified, with manual mixing after 4 min, and then terminated at 65 °C for 10 min. Detection was performed using a dipstick (KB-AMP-2T1C-S; K-BIOSCIENCES) and three drops (approx. vol. 100 µL) of DNA running buffer (K-BIOSCIENCES), and developed at room temperature for 15 min. A visible control line indicated successful reaction conditions. Some samples were also analyzed via 2.0% (w/v) agarose gel electrophoresis stained with Red safe (Vivantis, Selangor, Malaysia) and in Tris-borate-EDTA buffer (TBE). All the reactions were performed in duplicate or triplicate sets.

The reverse transcriptase recombinase polymerase amplification (RT-RPA) protocol was modified from Bonnet, van Jaarsveldt & Burt (2022). The reaction mixture contained: 29.5 µL rehydration buffer, 6.8 µL Nuclease-Free water, 2.1 µL FlaviPath-F probe (10 µM), 2.1 µL FlaviPath-R Probe (10 µM), 1 µL M-MLV Reverse Transcriptase (Invitrogen™, USA), and 1 µL RNase OUT™ (Invitrogen). The reaction mixture was divided into four aliquots of 10.6 µL each. After addition of 1 µL RNA template or 1 µL NF for NTC control and 0.29 µL MgAcO solution, reactions were incubated at 37 °C for 20 min and terminated at 65 °C for 10 min. Products were visualized using the dipstick or gel electrophoresis.

Dipstick results were scanned using a Canon MF3010 Printer and Scanner, and analyzed with ImageJ software (v1.54 g, NIH, USA). The mean gray intensity of the test and control bands were normalized by subtracting background intensity (averaged of upper and lower positions), and the T/C ratio was calculated (Ahmed et al., 2022). A sample was considered positive if its T/C ratio was more than three times that of the NTC (cutoff value). Slight negative values from low signals were set to zero. Sensitivity was calculated as: true positives/(true positives + false negatives), and specificity as: true negatives/(true negatives + false positives). The RT-RPA-LFD assay was performed on mosquito RNA samples (30–100 ng/µL) or diluted dengue RNA samples (100–104 copies) to determine the detection limit.

Reverse transcription quantitative polymerase chain reaction (RT-qPCR)

RT-qPCR was performed using a Stratagene Mxpro3000P system (Agilent Technologies, USA) with the QuantiNova SYBR Green RT-PCR Kit (Qiagen, Hilden, Germany). The 10 µL reaction included: 5 µL 2x Master Mix, 2.9 µL RNase-Free Water, 0.1 µL 100x RT Mix, 0.5 µL 10 µM forward primer, 0.5 µL 10 µM reverse primer, and 1 µL RNA Template. Primers DENV-2 NS5-F and DENV-2 NS5-R (Richardson et al., 2006) were used for dengue virus quantification; Zika virus quantification used primers ZIKF and ZIKR (Han et al., 2018) (Table S1). Cycling conditions included reverse transcription at 50 °C for 30 min, denaturation at 94 °C for 2 min, followed by 40 cycles of 94 °C for 15 s, 60 °C for 1 min, and 68 °C for 1 min, with a subsequent melt-peak analysis. To compare the detection efficiency of the qPCR and RPA-LFD assays, the same thawed RNA aliquot was used for both tests and analyzed on the same day without blinding. Results were accepted if qPCR efficiency ranged from 80–120%, with correlation coefficients (r2) between 0.9–1. The limit of quantification (LOQ) was defined as the lowest concentration within the linear dynamic range. No-template controls (NTC) were included in all analyses. A Cq of NTC greater than 40 (no Cq) or no band was observed at the target location in gel electrophoresis was considered free from target contamination.

General PCR conditions used final volumes of 10–25 µL, with products visualized on 2.0% (w/v) agarose gels. The amplifications were performed according to the following parameters: 1 cycle of 2 min at 95 °C, 35 cycles of 30 s at 95 °C, 1 min at 55 °C, and 1 min at 72 °C, followed by 1 cycle of 5 min at 72 °C. DNA concentrations were measured using a NanoDrop One Microvolume UV-Vis Spectrophotometer (Thermo Fisher Scientific).

Statistical analysis

Statistical analysis was performed using R version 4.3.3 (2024-02-29) on platform: aarch64-apple-darwin20 (64-bit) (R Core Team, 2024). Data normality was tested using Shapiro–Wilk test. Non-parametric data were analyzed using the Kruskal-Wallis test with post hoc comparisons using the ‘dunn.test’ and FSA packages (Dinno, 2024; Ogle et al., 2023). Student’s t-test was applied for comparisons between two datasets. A p-value of less than 0.05 was considered statistically significant.

Results

RPA primer design for pathogenic orthoflavivirus detection in mosquitoes

Primers targeting various orthoflaviviruses reported in previous studies were compiled (Table S3). Among the commonly targeted genes—Nonstructural Protein 5 (NS5), Nonstructural Protein 3 (NS3), and the Envelope protein (E) genes—NS5 was the most frequently used. As NS5 was the largest non-structural protein and conserved across different flavivirus species, it is a favorable target for molecular assays designed to detect members of this genus. Therefore, this study focused on broad-specificity primers targeting NS5.

Fourteen primer pairs from the literature were evaluated (Table S3). A set of 61 representative orthoflavivirus genome sequences—including 45 vertebrate-pathogenic and 16 non-vertebrate-pathogenic species—was retrieved from the ICTV and NCBI databases (Table S4). The binding potential of the 14 primer pairs to pathogenic and non-pathogenic orthoflaviviruses was compared (Table 1). Primer pairs that bound to >70% of pathogenic viruses were selected which included F8276d-F (Flav100F), pathogens 77.8% and non-pathogens 6.3%: F9063d-R (Flav200R), pathogens 100% and non-pathogens 87.5%: VD8, pathogens 84.4% and non-pathogens 12.5%: XF-F1, pathogens 88.9% and non-pathogens 25.0%: XF-F2, pathogens 75.6% and non-pathogens 37.5%: XF-R, pathogens 75.6% and non-pathogens 25.0%: YF1, pathogens 84.4% and non-pathogens 12.5%. However, as shown in Table S4, the genome sequences used for primer design in this study were derived from virus isolates collected between 1927 and 2016. Given the high mutation rates of RNA viruses, the effectiveness of the primers may be reduced against more recently emerging strains.

Table 1 In-silico analysis of broad specificity orthoflavivirus primers, including those from previous studies and newly designed in this study, was conducted against representative pathogenic and non-pathogenic orthoflaviviruses.

Primer	Detectable pathogen	No. pathogen (%)	Detectable non-pathogen	No. non-pathogen (%)	
cFD2	–	0 (0)	LamVba	0 (0)	
EMF1	DENVd, ZIKVj, ZIKVh, ZIKVi, JEVVm, WNVo, WNVq, YFVk, WSLVr, WSLVs	10 (22.2)	NAKVbc	2 (12.5)	
F2225-R	ZIKVh, ZIKVi, WNVp	3 (6.7)	–	0 (0)	
F5015-F	KOKVal	1 (2.2)	CYVbk	1 (6.3)	
F5807-R	DTMUVv, TMUVw	2 (4.4)	–		
F8276d-F (Flav100F)	DENVf, DENVa, DENVe, DENV DENVb, DENVg, DENVc, DENVd, ZIKVj, ZIKVh, ZIKVi, WNVo, YFVk, WSLr, WSLs, DTMUVv, TMUVw, SLEz, CPCVu, ILHVac, KUNVad, LGTVae, LIVag, EHVaj, SEPVak, KOKVal, JUGVam, SABVan, POTVao, BANVaq, UGSVar, KADVai, TBEaf, WNVq, WNVp, YFVl	35 (77.8)	RFVbe	1 (6.3)	
F9063d-R (Flav200R)	DENVf, DENVa, DENVe, DENVb, DENVg, DENVc, DENVd, ZIKVj, ZIKVh, JEVm, JEVn, ZIKVi, WNVo, WNVq, WNVp, YFVl, YFVk, WSLr, WSLs, DTMUVv, TMUVw, USUVx, USUVy, SLEz, BAGVaa, NTAVab, CPCVu, ILHVac, KUNVad, MVEVt, LGTVae, TBEaf, LIVag, BSQVah, KADVai, EHVaj, SEPVak, KOKVal, JUGVam, SABVan, POTVao, BOUVap, BANVaq, UGSVar, KEDVas	45 (100)	LamVba, LamVbb, CYVbk, AEFVbd, RFVbe, KRVbf, CXthFVbg, QBVbh, PCVbi CLBOVbi, CYVbk, CHAOVbl DGVbm, NOUVbn	14 (87.5)	
Flavi all AS4	DENVd, WNVo, WNVq, WNVp, USUVy, SLEz, TBEaf, SABVan POTVao, BOUVap, BANVaq	11 (24.4)	LamVba CYVbkCHAOVbl	3 (18.8)	
Flavi all S2	WSLr, NTAVab	2 (4.4)	NOUVbn	1 (6.3)	
Flavi-For	DENVf, DENVa, DENVe,DENVb, DENVg, DENVd, ZIKVj, JEVm, JEVn, WNVo, WNVq, WNVp, WSLr, USUVx, USUVy, SLEz, NTAVab, CPCVu, ILHVac, MVEVt, LGTVae, TBEaf, LIVag, KADVai, EHVaj, SEPVak, JUGVam, SABVan, POTVao, BANVaq	30 (66.7)	RFVbe, CYVbk, CHAOVbl, DGVbm	4 (25.0)	
FLAVI-NS5rev-1	WNVp	1 (2.2)	LamVba	1 (6.3)	
Flavi-Rev	DENVf, DENVa, DENVe, DENVb, DENVg, DENVd,ZIKVj, ZIKVh, JEVn, ZIKVi, WNVq, WNVp, YFVl, YFVk, WSLr, WSLs, DTMUVv, TMUVw,USUVx, USUVy, SLEz, BAGVaa, NTAVab, ILHVac, LGTVae, TBEaf,LIVag, BSQVah, KADVai, EHVaj, SEPVak, JUGVam, SABVan, POTVao, BOUVap, BANVaq, KEDVas	37 (82.2)	RFVbe, CYVbk, CHAOVbl, DGVbm, NOUVbn	5 (31.3)	
FlaviF1	DENVe, DENVb, DENVg, JEVm, JEVn, WNVo, WNVq, WNVp, YFVl, YFVk, SLEz, BAGVaa, NTAVab, ILHVac, LGTVae, TBEaf, LIVag, KADVai, JUGVam, SABVan, POTVao, BOUVap, BANVaq, UGSVar	24 (53.3)	–	0 (0)	
FlaviR2	DENVf, DENVa, DENVe, DENVb, DENVg, ZIKVj, ZIKVh, JEVm, JEVn, ZIKVi, WNVo, WNVq, WNVp, YFVl, YFVk, BAGVaa, CPCVu, KUNVad, LGTVae, KADVai, EHVaj, JUGVam, BOUVap, BANVaq, KEDVas	25 (55.6)	BARVbo	1 (6.25)	
MA	DENVe, DENVb, DENVg, ZIKVj, ZIKVh, JEVm, JEVn, ZIKVi, WNVq, YFVk, BAGVaa, CPCVu, JUGVam, BOUVap, BANVaq	15 (33.3)	BARVbo	1 (6.3)	
MAMD-F	DENVf, DENVa, DENVe,DENVb, DENVg, ZIKVj, ZIKVh, JEVm, JEVn, ZIKVi, WNVo, WNVq, WNVp, YFVl, YFVk, BAGVaa, CPCVu,KUNVad, LGTVae, KADVai, EHVaj, JUGVam, BOUVap, BANVaq, KEDVas	25 (55.6)	BARVbo	1 (6.3)	
Pan-flavivirus FW	ILHVac, LGTVae, LIVag, EHVaj, SEPVak,JUGVam, SABVan, KEDVas	8 (17.8)	AEFVbd	1 (6.3)	
Pan-flavivirus RV	DENVe, DENVb, DENVg, ZIKVh, JEVm, JEVn, ZIKVi, YFVk, WSLr, WSLs, NTAVab, KUNVad, MVEVt, TBEaf, LIVag	15 (33.3)	NOUVbn	1 (6.3)	
PFlav-fAAR	DENVf, WNVo, WNVq, WNVp, EHVaj	5 (11.1)	–	0 (0.0)	
PFlav-rKR	DENVf, WNVp, WSLs, USUVy, SLEz, TBEaf, SEPVak	7 (15.6)	CYVbk CHAOVbl, NOUVbn	3 (18.8)	
VD8	DENVf, DENVa, DENVe, DENVb, DENVg, DENVd, JEVm, JEVn, WNVo, WNVq, WNVp, YFVk, WSLr, WSLs, TMUVw, TMUVw, USUVx, USUVy, SLEz, BAGVaa, NTAVab, ILHVac, MVEVt, BSQVah, SEPVak, KOKVal	38 (84.4)	NOUVbn BARVbo	2 (12.5)	
XF-F1	DENVf, DENVa, DENVe, DENVb, DENVg, DENVc, DENVd, ZIKVj, ZIKVh, JEVm, JEVn, ZIKVi, WNVo, WNVq, WNVp, YFVj, YFVk, WSLr, WSLs, DTMUVv, TMUVw, USUVx, USUVy, SLEz, BAGVaa, CPCVu, ILHVac, MVEVt, LGTVae, TBEaf, KADVai, EHVaj, SEPVak, KOKVal, JUGVam, SABVan, POTVao, BOUVap BANVaq, KEDVas	40 (88.9)	RFVbe, CYVbk, CHAOVbl, BARVbo	4 (25.0)	
XF-F2	DENVf, DENVa, DENVe,DENVb, DENVg, DENVc, DENVd, ZIKVh, JEVm, JEVn, ZIKVi, WNVo, WNVq, WNVp, YFVk, WSLr, USUVx, SLEz, BAGVaa, ILHVac, KUNVad, MVEVt, LGTVae,TBEaf, LIVag, BSQVah, KADVai, SEPVak, KOKVal, JUGVam, BOUVap, BANVaq, UGSVar, KEDVas	34 (75.6)	RFVbe, CYVbk, CHAOVbl DGVbm, NOUVbnBARVbo	6 (37.5)	
XF-R	DENVf, DENVa, DENVe, DENVb, DENVg, DENVc, DENVd, ZIKVj, ZIKVh, JEVm, JEVn, ZIKVi, WNVq, WNVp, YFVk, WSLs, USUVy, SLEz, CPCVu, ILHVac, KUNVad, MVEVt, TBEaf,LIVag, BSQVah, KADVai, EHVaj, SEPVak, JUGVam, SABVan, POTVao, BOUVap, BANVaq, KEDVas	34 (75.6)	LamVba, CYVbk, CHAOVbl, DGVbm	4 (25.0)	
YF1	DENVf, DENVa, DENVe, DENVb, DENVg, DENVd, JEVm, JEVn, JEVn, WNVo, WNVq, WNVp, YFVk, WSLr, WSLs, TMUVw, USUVx, USUVy, SLEz, BAGVaa, NTAVab, ILHVac, ILHVac, MVEVt, BSQVah, SEPVak, KOKVal	38 (84.4)	NOUVbn, BARVbo	2 (12.5)	
YF3	JEVn, YFVk	2 (4.4)	–	0 (0.0)	
Flavi Path-F	DENVa, DENVg, DENVb, DENVe, DENVg, DENVc, DENVd, ZIKVh, ZIKVi, ZIKVj, YFVk, JEVVm, JEVVn, WNVo, WNVp, WNVq, WSLVr, MVEVt, DTMUVv, TMUVw, USUVx, USUVy, SLEVz, BAGVaa, NTAVab, ILHVac, KUNVad, LGTVae, TBEVaf, LIVag, BSQVah, KADVai, EHVaj, SEPVak, KOKVal, JUGVam, SABVan, POTVao,BOUVap, BANVaq, UGSVar, KEDVas	42 (93.3)	RFVbe, CHAOVbk, CHAOVbl, DGVbm, NOUVbn	5 (31.3)	
Flavi Path-R	DENVa, DENVg, DENVb, DENVe, DENVg, DENVc, DENVd, ZIKVh, ZIKVi, YFVk, YFVl, JEVVm, JEVVn, WNVo, WNVq, WSLVr, MVEVt, CPCVu	18 (40.0)	–	–	
FlaviPath-F and FlaviPath-R	DENVa, DENVg, DENVb, DENVe, DENVg, DENVc, DENVd, ZIKVh, ZIKVi, YFVk, JEVVm, JEVVn, WNVo, WNVq, WSLVr, MVEVt	18 (35.6)	–	–	
Notes.

DENV Dengue virus

ZIKV Zika virus

YFV Yellow fever virus

JEVV Japanese encephalitis virus

WNV West Nile virus

WSLV Wesselsbron virus

MVEV Murray Valley encephalitis virus

CPCV Cacipacore virus

DTMUV Duck Tembusu virus

TMUV Tembusu virus

USUV Usutu virus

SLEV St. Louis encephalitis virus

BAGV Bagaza virus

NTAV Ntaya virus

ILHV Ilheus virus

KUNV Kunjin virus

LGTV Langat virus

TBEV Tick-borne encephalitis virus

LIV Louping ill virus

BSQV Bussuquara virus

KADV Kadam virus

EHV Edge Hill virus

SEPV Sepik virus

KOKV Kokobera virus

JUGV Jugra virus

SABV Saboya virus

POTV Potiskum virus

BOUV Bouboui virus

BANV Banzi virus

UGSV Uganda S virus

KEDV Kedougou virus

LamV Lammi virus

NAKV Nakiwogo virus

AeFV Aedes flavivirus

RFV Royal Farm virus

KRV Kamiti River virus

CTFV Culex theileri flavivirus

QBV Quang Binh virus

PCV Palm Creek virus

CLBOV Calbertado virus

CHAOV Chaoyang virus

DGV Donggang virus

NOUV Nounane virus

BJV Barkedji virus

HGV Hepatitis G virus

The superscript denotes the virus acronym detailed in Table S4.

Although F9063d-R (Flav200R) showed complete binding to the selected pathogenic group, its high cross-reactivity (87.5%) with non-pathogens led to its exclusion from further consideration. Considering the optimal amplicon length for the TwistAmp® RPA kit (80–500 bp) (TwistDx Limited, 2023), only three primer pairs—XF-F1/XF-R (264 bp), XF-F2/XF-R (215 bp), and YF1/XF-R (326 bp)—met this requirement.

RPA primer design criteria included a length of 30–36 bp, GC content of 20–70%, melting temperature (Tm) of 50–100 °C, and fewer than five mononucleotide repeats. Among the forward primers (Table S5), YF1 had a relatively low Tm (50.3 °C), and XF-F1 had higher base repetition (4) and lower GC content (39.9%) than XF-F2 which could affect the binding affinity with nucleotide targets. Thus, XF-F2 and XF-R were selected for further development.

To optimize the assay in a cost-effective manner, a synthesized double-stranded DNA oligonucleotide was used as the RPA template instead of RNA or in vitro-transcribed RNA. Given the 130 bp maximum length for custom oligonucleotide synthesis and the ideal RPA amplicon range of 100–200 bp (TwistDx Limited, 2023), a new primer was designed to complement XF-F2. XF-F2 had a better GC content (50.8%) than XF-R (60.3%) and was selected for modification. The newly designed forward primer, FlaviPath-F, was derived from XF-F2 by incorporating degenerative bases into a highly similar region across several sequences in order to broaden its detection scope in pathogenic orthoflaviviruses, and then the primer length extended from 22 to 30 bp. The reverse primer, FlaviPath-R, was designed approximately 130 bp downstream from the FlaviPath-F, targeting conserved regions among major pathogenic orthoflaviviruses.

Binding ability of the FlaviPath primers is shown in Fig. 1 and Table 1. FlaviPath-F matched 93.3% of pathogenic and 31.3% of non-pathogenic orthoflaviviruses. FlaviPath-R matched 40.0% of pathogens and none of the non-pathogens. While XF-R had higher coverage, FlaviPath-R was more specific to key mosquito-borne pathogens, including DENV, ZIKV, YFV, JEV, WNV, Wesselsbron virus (WSLV), and Murray Valley encephalitis virus (MVEV).

Figure 1 The binding regions of primers Flavipath-F and Flavipath-R in the NS5 gene of representative pathogenic orthoflaviviruses.

Establishment of RPA-LFD detection method

The RPA-LFD assay (Figs. 2A1–2A2) was first tested using synthetic single- and double-stranded DNA oligonucleotides of dengue virus serotype 2 from Thailand (Den2Th, Table S2, Fig. S1), following the manufacturer’s protocol (39 °C for 20 min). Clear test lines were observed for double-stranded DNA samples (normalized intensity: 0.20–1.1, Table S6), whereas only weak signals were seen with single-stranded templates (normalized intensity 0.02−0.08). The difference from the no-template control (NTC) was statistically significant for double-stranded DNA (p = 0.01). Therefore, double-stranded oligonucleotides were used for subsequent testing.

Figure 2 Development of the RPA-LFD technique for pathogenic orthoflavivirus detection.

(A1-2) RPA reactions with synthesized single-stranded and double-stranded DNA oligonucleotide targets. (B1-2) RPA amplification with varying incubation times. (C1-2) RPA amplification at different incubation temperatures. Normalized intensity was calculated by subtracting the background intensity from the test line mean intensity and dividing it by the control line intensity. The cutoff value was set at three times the no-template control (NTC) intensity. Different lowercase letters indicate statistically significant differences among group means (P < 0.05). Error bars represent the standard error (SE) from duplicate or triplicate tests. Raw normalized intensity data are provided in Table S6. The electrophoresis gel image of the PCR product amplified from the Den2Th oligonucleotide template is shown in Fig. S1.

To optimize reaction time, the RPA-LFD assay was performed for 5, 10, 15, 20, and 25 min with the Den2Th double-stranded DNA oligonucleotide (Figs. 2B1–2B2). Amplification was observed at all time points (averaged normalized intensity 0.06–0.13), with the best signal at 20 min. Though the differences were not statistically significant across time points (except compared to NTC, p = 0.04), 20 min was chosen for future reactions.

The effect of incubation temperature was also evaluated (Figs. 2C1–2C2). Positive results were seen across all tested temperatures between 20 and 37 °C (normalized intensity 0.09–0.44). Statistically significant differences from the NTC were observed at all tested temperatures within the range (p ≤ 0.04). The strongest signal (average normalized intensity: 0.30) occurred at 37 °C, which was selected as the optimal reaction temperature.

Cross-reactivity evaluation

To assess cross-reactivity, synthetic double-stranded DNA oligonucleotides representing DENV, ZIKV, JEV, WNV, and three insect-only orthoflaviviruses (AEFV, NAKV, and LAMV) were synthesized (Table S2) and tested in the RPA-LFD assay (Figs. 3A1–3A2). Positive test results were observed for DENV, JEV, ZIKV, and WNV (normalized intensity 0.18–0.69), while AEFV, NAKV, and LAMV showed no signal above the cutoff threshold of 0.03 (normalized intensity 0.00–0.02). These results matched primer predictions (Table 1).

Figure 3 Evaluation of RPA-LFD cross-reactivity detection for multiple viruses.

(A1-2) RPA-LFD detection of synthesized oligonucleotide DNA templates from multiple orthoflavivirus targets. (B1-2) RT-RPA-LFD detection using RNA extracts from dengue- and Zika-infected mosquitoes (samples A, B, C, and D). RT-qPCR analysis results for these mosquito samples are shown in Figs. S2 and S3. Neg represents RNA samples from uninfected mosquitoes. Normalized intensity was calculated as described for Fig. 2. Different lowercase letters indicate statistically significant differences among group means (P < 0.05). Error bars represent SE from duplicate or triplicate tests. Raw normalized intensity data are shown in Table S6.

RT-RPA was then conducted on RNA extracted DENV serotype 2- and ZIKV-infected mosquitoes (Figs. 3B1–3B2). Infection status was confirmed by RT-qPCR (Figs. S2 and S3). All virus-infected mosquito RNA samples (A–D) produced a clear RT-RPA signals using FlaviPath-F and FlaviPath-R (DENV: 0.23–0.60, ZIKV: 0.12–1.11). In contrast, uninfected laboratory colony samples (Neg 1 and 2) and NTCs showed negligible signal (≤ 0.01). These results confirmed the effectiveness of the RT-RPA-LFD assay in distinguishing infected from uninfected mosquito samples.

Detection limit determination

The detection limit of the RPA-LFD test for orthoflaviviruses was evaluated using a serial dilution technique with the RNA extracted from dengue-infected mosquito samples. These samples (A, B, and C) contained dengue virus at densities of 105, 104, and 106 RNA copies, respectively (Figs. 4A and 4B), as quantified by RT-qPCR using an absolute quantification method with in vitro transcribed dengue serotype 2 RNA as a standard (Fig. S2). Sample A was serially diluted tenfold down to a single gene copy.

Figure 4 Sensitivity of RT-RPA-LFD for detecting dengue virus in mosquito samples.

(A–B) RT- RPA-LFD detection of serial 10-fold dilutions of RNA from DENV-2-infected mosquitoes (sample A, B, and C, Fig. S2). (C–D) RT-qPCR analysis of the same mosquito RNA samples and gel electrophoresis of RT-qPCR products amplified with DENV-2 NS5-F and DENV-2 NS5-R primers (177 bp amplicon). Normalized intensity and cutoff criteria were applied as in previous figures. Different lowercase letters indicate statistically significant differences among group means (P < 0.05). Error bars represent SE from multiple tests. Raw normalized intensity data are provided in Table S6.

Positive RT-RPA-LFD results were obtained across all tested dilutions (101–104 RNA copies), though with varying detection rates (60–100%). Mosquito samples B and C also yielded positive RPA results in the RT-RPA-LFD assay (Figs. 4A and 4B). In comparison, RT-qPCR consistently detected dengue virus only at concentrations above 102 RNA copies (Figs. 4C and 4D). While the RT-RPA-LFD achieved 88.6% sensitivity (5 false negatives out of 44 samples, Figs. 3–4) and 89% specificity (2 false positives out of 24 samples, Figs. 2–4) in the range of 101–104 RNA copies, RT-qPCR demonstrated 100% sensitivity and specificity within the tested range of 102–104 RNA copies. The agreement between RT-RPA-LFD and RT-qPCR, measured by Cohen’s Kappa, was 0.6 (standard error: 0.17), indicating substantial agreement (Landis & Koch, 1977).

Although RT-RPA-LFD demonstrated lower reliability compared to RT-qPCR analysis, it provided faster processing and greater practical utility, particularly in field settings. These findings suggest that RT-RPA-LFD is suitable for detecting low viral loads or pooled samples, however, caution is advised due to the risk of false negative results at concentrations below 104 RNA copies. The estimated limit of detection for the RT-RPA-LFD assay was 104 RNA copies, as not all positive mosquito samples were consistently detected below this threshold.

Discussion

In this study, we developed novel RPA primers that, based on the in-silico analysis of their NS5 genes, potentially detected dengue virus (DENV), Zika virus (ZIKV), yellow fever virus (YFV), Japanese encephalitis virus (JEV), West Nile virus (WNV), Wesselsbron virus (WSLV), and Murray Valley encephalitis virus (MVEV). We demonstrated the utility of our RPA-LFD assay by detecting oligonucleotides of DENV, ZIKV, JEV, and WNV, as well as DENV- and ZIKV-infected mosquito samples. This assay offers a practical tool for estimating viral infection rates in wild mosquito populations, serving as an early warning system to inform timely vector control strategies and mitigate public health threats.

A literature review revealed limited previous work using RPA to detect diverse orthoflaviviruses. Bonnet, van Jaarsveldt & Burt (2022) developed two sets of RPA probes with LFD systems for WNV and WSLV detection with sensitivities of 1.9 ×101 and 3.5 ×100 copies, respectively, using in vitro transcribed RNA. Myhrvold et al. (2018) showcased the Cas13-based SHERLOCK (specific high-sensitivity enzymatic reporter unlocking) platform, detecting ZIKV (down to one copy) and DENV (10–100 copies depending on serotype) with serotypes-specific CRISPR RNAs. Ma et al. (2023) introduced Flap endonuclease 1-aided RPA (FARPA) to detect ZIKV, tick-borne encephalitis virus, YFV, and Chikungunya virus, achieving detection limits as low as 20 copies. In comparison, our RT-RPA-LFD test reliably detected DENV serotype 2 in infected mosquitoes reliably at concentrations above 104 copies, although detection was still possible in most cases at concentrations as low as a single copy. While the sensitivity of our assay was slightly lower than that of previously developed advanced assays, this is notably the first study to apply RPA for detecting a broad range of orthoflaviviruses in mosquito vectors.

The use of degenerated bases in the RPA primers may reduce assay sensitivity. Previous studies (Bonnet, van Jaarsveldt & Burt, 2022; Myhrvold et al., 2018) incorporated only 2–4 degenerate bases per RPA primer. In contrast, our FlaviPath forward and reverse primers contained nine and 14 degenerate bases, respectively. A high number of degenerate bases results in a mixture of primer variants, each representing a different base combination. When the same total primer volume is added to the RPA reaction, a greater degree of degeneracy means fewer molecules of each specific primer variant are present. In reactions with low template concentrations, the likelihood that a fully complementary primer will encounter and bind to the target sequence is reduced, leading to lower detection sensitivity. Therefore, minimizing the number of degenerate bases in primer design would likely enhance sensitivity.

Our RT-RPA-LFD method demonstrated a detection efficacy of 60–100% across a concentration range of 1–104 RNA copies, whereas RT-qPCR consistently maintained 100% sensitivity from 102 to 104 copies. A similar trend of reduced sensitivity at very low concentrations has been reported previously (Panpru et al., 2021). Therefore, caution is advised for samples containing fewer than 103 RNA copies due to the risk of false-negative results. However, as shown in Fig. S2, the viral load in whole body mosquitoes often exceeds 104 copies, which falls within the reliable detection range of our assay. This is further supported by a previous study that quantified dengue virus titers in the salivary glands of field-caught mosquitoes, reporting concentrations ranging from log 5.23 to 7.26 copies (Moura et al., 2015). Additional studies using artificial infectious blood feeding reported post-incubation viral titers after 7–14 days within the following ranges: whole-body DENV titers of log 5.47–6.19 (Conceição, Da Poian & Sorgine, 2010) and log −1.5 to 4.9 (Yang et al., 2010); DENV titers in the midgut of log 5–8 (Richardson et al., 2006) and log 3.27–4.69 (Wu et al., 2022); and DENV in salivary glands ranging log 3–6 (Raquin & Lambrechts, 2017). Comparable midgut titers have also been reported for other arboviruses, including Usutu virus (USUV) at log 8.37–log 8.85, WNV at log 9.60–10.10, and ZIKV at log 8.11–10.03 copies (Tang et al., 2020). These findings support the suitability of our RT-RPA-LFD assay for large-scale field surveillance, with RT-qPCR serving as a confirmatory method in focused or low-prevalence settings.

Previous RPA test for DENV alone in mosquitoes achieved 92–100% sensitivity across serotypes (Ahmed et al., 2022). Our broadly targeting design showed slightly lower sensitivity (88%), reflecting the trade-off between broad detection and specificity. However, this broader approach optimizes resource use by identifying low-risk areas, reserving pathogen-specific assays for high-risk regions during outbreakes, reducing overall surveillance costs.

Given the rapid evolution of viruses, perfectly conserved genomic regions are difficult to identify. As our research strategy was to synthesize DNA oligonucleotides that could be used as a target for the test development, we had to limit the length of the amplified amplicon to the most conserved region, less than 130 bp, and ensuring the detection of major mosquito-borne pathogenic orthoflaviviruses. Future studies might consider leverage longer synthetic gene fragment technology, that can prepare DNA sequences from 125 bp to several thousand bases, enabling more extensive regions for searching the better-conserved regions for probe binding sites.

Our in-silico analysis also highlighted previously published primers with broad binding potential. The XF-R primer (Xue et al., 2021) could recognize a wide range of pathogenic orthoflaviviruses, including tick-borne encephalitis virus which is an important emerging virus. Likewise, F9063d-R (Flav200R) (Grubaugh et al., 2013; Maher-Sturgess et al., 2008) could target both pathogenic and non-pathogenic orthoflaviviruses. These primers may aid in the discovery of novel orthoflaviviruses across vectors such as mosquitoes, ticks, and bats.

Comparing Table 1 and Table S4, we noted that while our FlaviPath-F and FlaviPath-R primers successfully matched to all selected DENV, JEV, and WNV strains, they did not bind to all strains of ZIKV, YFV, or WSLV, suggesting that future primer optimization may be necessary to enhance detection across diverse viral strains.

RNA viruses generally exhibit high mutation rates, typically ranging from 10−3 to 10−5 mutations per nucleotide per replication cycle (mut/nt/rep), with most falling around 10−3 (Duffy, Shackelton & Holmes, 2008). This rate is significantly faster than that of eukaryotes, which ranges from 10−7 to 10−11 mutations per nucleotide per generation (Bergeron et al., 2023). The rapid mutation in RNA viruses is primarily due to the low fidelity of their RNA-dependent RNA polymerases (RdRps), along with contributions from host antiviral enzymes, spontaneous chemical reactions, and ultraviolet radiation (Duffy, Shackelton & Holmes, 2008). The genome sequences used for primer design in this study were derived from virus cultures collected between 1927 and 2016, as detailed in Table S4. Given the fast-paced evolution of RNA viruses, our RPA primers may have reduced sensitivity to newly emerged strains and may require future modification.

The large error bar observed in the lateral flow dipstick results may partially reflect manual inconsistencies in dispensing the DNA running buffer. Variability in drop size during manual application could slightly alter the reaction mixture’s final concentration. Although precise pipetting could improve reproducibility, we prioritized a simplified field-adaptable method using dropper bottles over pipettes.

Occasionally, faint test bands were observed in the negative or no-template controls (NTCs). This was also reported in previous RPA studies (Myhrvold et al., 2018; Patchsung et al., 2020; Thanakiatkrai, Chenphun & Kitpipit, 2024; Zhang et al., 2020). To mitigate false positives, we applied a cutoff intensity threshold threefold greater than the NTC control, following Ahmed et al. (2022). Primer-dimer artifacts likely contributed to background signals (Zhang et al., 2020). While nfo endonuclease IV or exonuclease III-based RPA methods (Ahmed et al., 2022; Escadafal et al., 2014; TwistDx Limited, 2023) may minimize these. However, as our study focuses on a multiplex detection system targeting multiple viruses, we could not identify sufficiently conserved regions within the ∼130 bp amplicon to enable the use of these probe-based modifications. Future studies targeting specific viral sequences may benefit from incorporating these enhanced methods.

In addition, blind testing should be incorporated in future studies. Occasionally, we encountered unexpected results and repeated the experiments to verify that they were not caused by human or technical errors. However, this approach may introduce bias, potentially leading to an overestimation of assay efficiency. In real-world scenarios, where samples are truly unknown, the actual performance may be lower than what is observed under controlled laboratory conditions. Therefore, implementing blind testing would help eliminate subjective bias, leading to a more objective evaluation and increased reliability in estimating assay efficiency.

Developing a simple RNA preparation method remains a priority. Ahmed et al. (2022) and Pollak et al. (2023) used the TNA-Cifer Reagent (BioCifer, Auchenflower, Australia) and incubated on ice or at room temperature for 10 min for rapid RNA extraction in mosquito and blood samples, although it was not available in our region. Alternatively, Myhrvold et al. (2018) employed a simple chemical and heat treatment using tris(2-carboxyethyl)phosphine hydrochloride (TCEP) and EDTA at 37 °C for 20 min and 64 °C for 5 min for blood and saliva or at 95 °C for 10 min for urine samples. Future work should explore affordable, field-compatible RNA extraction techniques to further facilitate point-of-care viral detection.

Conclusions

This is the first to develop an RPA-based assay for detecting multiple pathogenic orthoflaviviruses in mosquito vectors. By adapting and modifying broad-specific primers for the Orthoflavivirus genus, we created the FlaviPath-F and FlaviPath-R primers and established a corresponding RT-RPA-LFD detection method. Our newly designed primers potentially detect DENV, ZIKV, YFV, JEV, WNV, WSLV, and MVEV based on the in-silico NS5 genes analysis. The RPA-LFD test succesfully detected DENV, JEV, ZIKV, and WNV DNA oligonucleotide, as well as DENV and ZIKA in infected mosquitoes. It demonstrated high sensitivity with a detection limit of 104 RNA copies, which falls within the concentration range often observed in mosquitoes. This technique holds promise as an early warning platform for future arbovirus outbreaks, offering a practical tool for large-scale surveillance in both resource-limited and epidemic settings.

Supplemental Information

Supplemental Information 1 Supplementary Materials

Supplemental Information 2 MIQE checklist

The authors gratefully thank Assoc. Prof. Dr. Benchaporn Lertanantawong, Prof. Dr. Worachart Sirawaraporn, and Assoc. Prof. Philip D. Round of Mahidol University for their valuable comments and suggestions throughout this research.

Additional Information and Declarations

Competing Interests

Author Contributions

Ethics

Data Availability

The authors declare there are no competing interests.

Parinda Thayanukul conceived and designed the experiments, performed the experiments, analyzed the data, prepared figures and/or tables, authored or reviewed drafts of the article, and approved the final draft.

Ronald Enrique Morales Vargas conceived and designed the experiments, performed the experiments, authored or reviewed drafts of the article, and approved the final draft.

Konkamon Sujijun performed the experiments, analyzed the data, prepared figures and/or tables, and approved the final draft.

Pimchanok Khumpeera performed the experiments, analyzed the data, prepared figures and/or tables, and approved the final draft.

Kittiya Suksawat performed the experiments, analyzed the data, prepared figures and/or tables, and approved the final draft.

Nahallage Dona Asha Dilrukshi Wijegunawardana performed the experiments, analyzed the data, prepared figures and/or tables, and approved the final draft.

Patsamon Rijiravanich conceived and designed the experiments, authored or reviewed drafts of the article, and approved the final draft.

Werasak Surareungchai conceived and designed the experiments, authored or reviewed drafts of the article, and approved the final draft.

Pattamaporn Kittayapong conceived and designed the experiments, authored or reviewed drafts of the article, and approved the final draft.

The following information was supplied relating to ethical approvals (i.e., approving body and any reference numbers):

The use of mosquito specimens in this study was approved by the Faculty of Science, Mahidol University Animal Care and Use Committee (SCMU-ACUC) under protocol number MUSC66-020-650.

The following information was supplied regarding data availability:

The data are available in the Supplementary File.

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
