# Peer review of "Reverse transcription recombinase polymerase amplification-lateral flow assay for detection of pathogenic orthoflaviviruses in mosquito vectors"

_PeerJ, doi:10.7717/peerj.19820_

## Round 0.1 · original submission · Major Revisions

Dear authors, I recommend that you improve the manuscript in accordance with the reviewers' comments and completely revise the statistical processing of the data. You probably need to consult with a specialist in this field. Figures 2-4 require the use of multiple sample comparison methods (I recommend the Tukey test). Data (if there are more than 4 replicates) should preferably be presented as a box plot with median, first and third quartiles, and maximum and minimum values. These figures will probably be significantly reorganized. Please note that the size of numbers and letters in all figures should be approximately equal to the height of letters in the text of the article (8-9 points).

The description of statistical processing of data in the text of the article should be very detailed (add a separate last subsection to Material and Methods). Did you check the distribution of data in each of the samples for normality? What did you do if the sample had significant asymmetry or kurtosis?

I hope that the new version of your manuscript will be recommended for publication by all four reviewers.

Reviewer 1 ·

Basic reporting

Introduction
It is necessary to better mention how this amplification technique works and what advantages it has.
It is necessary to highlight the advantage that this methodology has over other isothermal PCRs, since several articles that have implemented other isothermal PCR strategies for the same viruses have implemented other isothermal PCR strategies for these viruses, and in this introduction it is necessary to highlight more the importance of the article and what advantages this test has compared to others.

Experimental design

Methodology
Lines 169, correct the capitalization
Describe the species of mosquitoes used, whether they were from fields or infected in laboratories, in the case of this article it should be better increased how the RNA was obtained, and the number of samples evaluated for each of the viruses

Make a more detailed legend for figure two, where the results shown in figure 2 are better described, even this observation is for all the legends of figures and tables, because they are missing. It is important to mention how the intensity or cut-off point is defined or calculated for each of the viruses, and it is also necessary to define how many negative samples are actually classified correctly.
It does not say what DNA dilutions are used to establish analytical sensitivity points for the technique, this is defined in the results, but it is worth mentioning it in the methodology
Many of the results are organized in supplementary tables, but some could be arranged in graphs in the article for a better understanding
I think it is important to clarify and better express in the results the concordance (Kapa index) with other tests and to specify in the methodology how the sensitivity and specificity were determined
The species of the mosquitoes used was not mentioned, whether they were infected in the laboratory or are field mosquitoes, this is mentioned in the results, but it should be specified in the methodology
All the legends of the graphs lack more details, for example, in figure 4 each gel does not appear clearly.

Validity of the findings

In my opinion, the impacts of the results are limited, I think that they can be emphasized more in the discussion.
The statistical analysis is not very clear.

·

Basic reporting

The manuscript "Reverse transcription recombinase polymerase amplification-lateral flow detection of pathogenic Orthoflaviviruses in mosquito vectors" described the use of RPA techonoque to detect various orthoflaviviruses in mosquito vectors. The language applied, literature reference, structure and relevant results are satisfatory.

Experimental design

The experimental design are well-done and well-described. I suggest to the authors to insert a figure to better ilustrate the detection of each pathogenic and non-pathogenic flavivirus with each specific primer.

Validity of the findings

The results are instersting and could be applied for other researchers in order to assay for pathogenic flaviviruses. I suggest to the authors to insert a figure to better ilustrate the detection of each pathogenic and non-pathogenic flavivirus with each specific primer.

Reviewer 3 ·

Basic reporting

Although the manuscript is readable, English language editing would improve the grammar (verb tense, sentence structure, plurality, etc.) and readability of the manuscript. Professional editing for clear, unambiguous, professional (scientific) English throughout is needed.

RNA integrity of nucleic acid extraction is deemed essential in the MIQE checklist and based on the requested RIN/RQI or RNA Electrophoresis traces (e.g. Agilent bioanalyzer results), the response of nanodrop and electrophoresis of cDNAs likely does not meet this criteria.

The literature references seem sufficient. The abstract would be enhanced by a brief justification on the utility of developing RPA methods compared to available PCR detection methods.

Minor comments:
There should be consistent use of abbreviations for viruses – once the abbreviation is defined at first use, continue to use this abbreviation in place of the virus name through the manuscript.

Lines 34-35: it would be good to pay attention to the subtle difference between the viruses and the disease they cause (e.g. Dengue virus is the causative agent of Dengue fever).

Line 41: “specification with” should probably be “specificity for” or “cross-reactivity with” – many additional examples where clear, unambiguous English could be enhanced are found through the manuscript.

The resolution of the graph in Figure 4C does not allow the reader to make out the axis labels.

Experimental design

Thayanukul et. al., describe the application of RPA for the detection of human pathogenic flaviviruses using a newly designed primer pair. While a case is made that this method is easier to run, especially in field surveillance of mosquito samples, the authors do not appear to have considered intra-strain genetic variation in this manuscript. It remains unclear if these are robust to naturally observed genetic variation within the viruses and strains tested.

The research question could use refinement to narrow down the intended use for the RPA assay.

The authors have mostly done an excellent job at describing their research in meticulous details in the methods. Some of this could probably be summed up and/or covered in the supplemental data (e.g. Lines 162-170, are standard for triZOL RNA extraction and could be summed up with: “RNA was isolated from triZOL according to the manufacturer’s instructions.”

Line 158-161: Please add the experimental conditions of mosquito infection to generate these reference samples. Including the DENV or ZIKV strain used for infections would help reproducibility. How was homogenization accomplished (bead based)?

Validity of the findings

In Figure 2 - A2, B2, and C2; Figure 3 – A2; and Figure 4 - B, the error bars appear very large and overlapping. Appropriate statistical testing should be conducted to verify that these assays can reliably distinguish differences in these parameters.

Figure 4A appears to demonstrate a major problem with these findings, it is difficult for this reviewer to visually identify the bands on many of the test-sticks from groups with reported 100% sensitivity. Additionally, it seems like there are clear bands in the 104 group where there should not be for specificity. Combined with the large error bars and lack of statistics, this would seem to call into question the validity of the findings.

Additional comments

Minor comments:
Line 105: Tick-Borne should be capitalized. TBEV/TBE are important emerging viruses that were not introduced among the other medically relevant Flaviviruses. If the focus of this manuscript is on mosquito-borne flaviviruses, this seems like a rational reason to not focus on the tick-borne viruses. Commentary from the authors on this topic in the introduction and/or discussion would be helpful.

Line 106: Slightly confusing to include CHIKV (an Alphavirus) in this sentence without pointing out that it is outside of Flaviridae.

Line 124-126: Please provide NCBI accession numbers for the sequences used for primer design – this does seem to be included in the supplementary, but calling it out here would be useful.

·

Basic reporting

The topic of the article under review is very relevant. Recombinase polymerase amplification is an alternative to the polymerase chain reaction, which has a number of advantages. This is an isothermal reaction that occurs at a low speed and at room temperature using inexpensive and simple equipment, compared to PCR. The obtained research results are important for the development of inexpensive, rapid, "field" molecular diagnostic tests for viral pathogens, including the presence of orthoflaviviruses in mosquito vectors. The research questions and hypotheses in the manuscript are defined. The results and conclusions are confirmed by statistical analysis. The manuscript is formatted according to the requirements. However, there are some minor comments.

Experimental design

Lines 1-3. I recommend shortening the title of the manuscript. The title of the article should be short, clear and concise.
Lines 63-68. The sentence is too long and difficult for readers to understand. I recommend breaking it into two parts. There are many similar sentences in the text of the manuscript.
Line 65. Add the word “cause” (hemorrhagic fever) to the phrase “…that damage the central nervous system and hemorrhagic fever…”.
Lines 79-83 require citation.
Lines 182-185. Rephrase the sentence, put the verbs in the past tense.
Lines 191-192. Rephrase the sentence. Indicate where the polymerase amplification protocol was modified from. For example: “…modified from the generally accepted method of conducting analyses (Bonnet et al., 2022)”.

Validity of the findings

Lines 367-373. I recommend moving the paragraph to the “Conclusions” section.

Additional comments

No comments.

---

## Round 0.2 · Major Revisions

Dear Dr. Thayanukul, I ask you to carefully analyze the reviewer's comments. The shortcomings pointed out by the reviewer are very significant. If possible, I ask you to conduct additional experiments that will remove suspicions of methodological errors in this study.

Reviewer 3 ·

Basic reporting

An additional readthrough for clear and unambiguous English would still be helpful.

Experimental design

The RNA quantification in figure S1 and S2 raise an important additional question. These are reported in the range of 16-68.1 ng/L as determined by NanoDrop. This is likely a conversion error as the maximum sensitivity of NanoDrop for RNA quantification is around 2 mg/L (2ng/µL). Please revise as needed.

Validity of the findings

The statistical analysis seems to highlight important limitations of RPA assay. First, it seems that the assay is highly sensitive to time with the 20-minute assay time yielding significant results, but the assay is robust to changes of plus-or-minus 5 minutes. Similarly, the assay seems to be temperature dependent with 25°C and 37°C yielding significant results but not 20°C, 30°C, or 42°C. In the absence of a biological explanation of why intermediate temperatures were not significantly different than negative controls, this raises a question about appropriate prospective design to achieve statistically reproducible results.

The statistical analysis of Figure 3 suggests only DENV and ZIKV can be distinguished from NTC as synthetic nucleotides, but that only ZIKV is distinguishable in mosquito samples. The abstract and discussion seem to suggest that the RPA assay would be useful for several Flaviviruses, but as a mosquito surveillance tool, these data suggests that only ZIKA is reliably detectable.
In Figure 4, the 10^1 sample has a large error bar and visually, the strips look different than the others. Given the sensitivity of the assay to time and temperature, this could suggest that it may be difficult to repeat the results for this sample. Excluding the 10^1 sample, it seems like there is not true detection until the 10^4 assay condition. Taken together, this might suggest that the true limit of detection of the RPA assay is 10^4.

Considering the statistical analysis, the conclusions seem overstated. I would suggest revising to better reflect the ability of the RPA assay to consistently detect Flaviviruses as supported by your data.

Additional comments

no comments

·

Basic reporting

The authors took into account all the comments I made. The manuscript can be recommended for publication.

Experimental design

The authors took into account all the comments I made. The manuscript can be recommended for publication.

Validity of the findings

The authors took into account all the comments I made. The manuscript can be recommended for publication.

Additional comments

No comments.

---

## Round 0.3 · Minor Revisions

Dear Dr. Thayanukul, I ask you to carefully correct the shortcomings pointed out by the reviewer before the article is accepted for publication.

Reviewer 3 ·

Basic reporting

no comment

Experimental design

no comment

Validity of the findings

no comment

Additional comments

The authors have appropriately addressed my previous concerns.

·

Basic reporting

This study aims to develop a novel sensitive and specific RT-RPA-LF detection system for the broad pathogenic orthoflaviviruses in mosquito vectors. The methodology is appropriately performed and described. I consider this research publishable only if the current comments are appropriately addressed.

• On line 74, could you describe the current global burden of the most important orthoflaviviruses in humans?

Experimental design

• On line 133, did you follow a particular inclusion criterion for retrieving the sequences? Did you aim for the more updated? Considering you are selecting primer sets according to the in silico matching, I would like to suggest you provide additional information regarding the sequences you are using, like the collection date.
• On line 140, could you cite the studies from which you took the primer sequences?
• On line 141, how did you determine the conserved region? Based on the literature? Did you perform further analysis?
• On line 154, could you please clarify if the synthetic oligos contained the T7 promoter sequence? If not, could you explain how you synthesize the RNA segments without the T7 promoter sequences in the DNA template?
• On line 219, could you please clarify whether you analyzed the samples the same day for RT-RPA and RT-qPCR? Were there any freeze-thaw cycles between the RT-RPA and RT-qPCR analyses? Did you perform the blinded test?
• On line 251, could you comment on the collection date or how recent the sequences were, which the primers showed more theoretical matching according to the in silico analysis?
• Did your RPA-LF utilize a probe? could you please show in Figure 1 the target sequence segment aligned by the RPA probe?
• On Figure 2, could you please introduce the evidence of an image of gel electrophoresis analysis showing the amplicons for the RPA testing of the DNA standards?
• On Figure 4, could you please introduce the evidence of gel electrophoresis analysis showing the amplicons of the RT-RPA testing the limit of detection using RNA samples?

Validity of the findings

• On LINE 364, could you try to explain why you obtained a higher limit of detection compared to previous RPA studies? What about the use of degenerate primers? Do you think this somehow may affect the analytical sensitivity?
• Could you identify and disclose some of the limitations of your study? For example, did you perform the blind test? For the primer selection against genetic sequences, did you compare against the more recent sequences? How useful may your primers be based on the current circulating orthoflaviviruses?
• On line 428, you mentioned the orthoflavivirus viral load in mosquitoes typically falls in the 104 copies. Could you cite in the introduction or discussion some studies exploring the viral load of orthoflavivirus viral load in mosquitoes to support your conclusion?

Additional comments

I found your research very interesting, and I hope my comments will be useful in improving the quality of the publication

---

## Round 0.4 · accepted · Accept

Dear Dr. Thayanukul, I congratulate you on the acceptance of this article for publication and believe that it will be a significant contribution to the issue you are studying. I wish you further success in your research into mosquito-borne viruses. This is a global, very important problem in practical public health.

·

Basic reporting

No comment

Experimental design

No comment

Validity of the findings

No comment

Additional comments

Dear authors, thank you very much, I believe you have appropriately addressed the point to point comments. .